# Preparation of S-Scheme g-C₃N₄/ZnO Heterojunction Composite for Highly Efficient Photocatalytic Destruction of Refractory Organic Pollutant

Buse Sert [1], Zeynep Bilici [2], Kasim Ocakoglu [1,*], Nadir Dizge [2], Tannaz Sadeghi Rad [3] and Alireza Khataee [3,4,*]

[1] Department of Engineering Fundamental Sciences, Faculty of Engineering, Tarsus University, 33400 Tarsus, Turkey
[2] Department of Environmental Engineering, Mersin University, 33343 Mersin, Turkey
[3] Department of Environmental Engineering, Faculty of Engineering, Gebze Technical University, 41400 Gebze, Turkey
[4] Research Laboratory of Advanced Water and Wastewater Treatment Processes, Department of Applied Chemistry, Faculty of Chemistry, University of Tabriz, Tabriz 51666-16471, Iran
* Correspondence: kasim.ocakoglu@tarsus.edu.tr (K.O.); akhataee@gtu.edu.tr (A.K.)

**Abstract:** In this study, graphitic carbon nitride (g-C₃N₄)-based ZnO heterostructure was synthesized using a facile calcination method with urea and zinc nitrate hexahydrate as the initiators. According to the scanning electron microscopic (SEM) images, spherical ZnO particles can be seen along the g-C₃N₄ nanosheets. Additionally, the X-ray diffraction (XRD) analysis reveals the successful synthesis of the g-C₃N₄/ZnO. The photocatalytic activity of the synthesized catalyst was tested for the decolorization of crystal violet (CV) as an organic refractory contaminant. The impacts of ZnO molar ratio, catalyst amount, CV concentration, and $H_2O_2$ concentration on CV degradation efficiency were investigated. The obtained outcomes conveyed that the ZnO molar ratio in the g-C₃N₄ played a prominent role in the degradation efficiency, in which the degradation efficiency reached 95.9% in the presence of 0.05 mmol of ZnO and 0.10 g/L of the catalyst in 10 mg/L of CV through 120 min under UV irradiation. Bare g-C₃N₄ was also tested for dye decolorization, and a 76.4% dye removal efficiency was obtained. The g-C₃N₄/ZnO was also tested for adsorption, and a 32.3% adsorption efficiency was obtained. Photocatalysis, in comparison to adsorption, had a dominant role in the decolorization of CV. Lastly, the results depicted no significant decrement in the CV degradation efficiency in the presence of the g-C₃N₄/ZnO photocatalyst after five consecutive runs.

**Keywords:** S-scheme photocatalyst; g-C₃N₄/ZnO; advanced oxidation processes; crystal violet

## 1. Introduction

Wastewaters from the textile industry have adverse impacts on aquatic life due to the high concentration of hazardous chemicals released into the environment [1]. Diverse salts, chlorinated chemicals, and surfactants are commonly found in such wastewaters [2]. Since most dyes own high reactivity, even a trace amount of dye in water affects the gas solubility in water bodies [3,4]. Hence, researchers have devised several treatment techniques for the disintegration of hazardous contaminants [5]. Amongst these techniques, the photocatalytic process as a commonly used method for wastewater treatment has gained high interest [6].

The photocatalytic process is considered an eco-friendly, low-cost, and effective alternative method for decreasing water pollution when compared to other chemical, physical, and biological approaches. [7,8]. By using light irradiation, electrons in the valence band (VB) of a semiconductor are stimulated to the conduction band (CB), generating electron–hole pairs [9,10]. Afterward, the formed electron–hole pairs are separated to trigger the desired redox reaction on the active catalyst surfaces [11]. Metal oxides, especially titanium dioxide (TiO₂) and zinc oxide (ZnO), are frequently utilized in photocatalytic investigations owing

to their physical stability and strong catalytic activity [12,13]. Moreover, ZnO has unique features, such as high abundance, low toxicity, ecofriendliness and cost-effectiveness [14]. ZnO is an n-type semiconductor oxide comparable to $TiO_2$. As ZnO has the same band gap energy as $TiO_2$ but exhibits greater absorption efficiency across a significant portion of the solar spectrum, ZnO has been suggested as a potential replacement photocatalyst [15–17]. Mirzaeifard et al. [18] reported that in the presence of 0.1 g of S-doped ZnO and 5 mg/L of RhB at a pH = 5, a degradation efficiency of 100% was achieved through 90 min. In addition, Ahmad et al. [19] prepared Au-ZnO which could degrade RhB (95%) within 60 min under the basic pH of RhB. Nevertheless, a high recombination ratio of electron hole pairs and photo corrosion have constrained the usage of ZnO as a photocatalyst [20].

Direct Z-scheme heterojunction and S-scheme heterojunction both use the same basic carrier separation mechanism. The first two generations of direct Z-scheme heterojunctions, conventional and all-solid-state, both have significant drawbacks. As a result, S-scheme heterojunction with a clear identification and a practical charge-transfer channel can successfully prevent needless misunderstandings (Figure 1). In addition, due to their distinctive structure and photoelectronic properties, such as greater specific surface area and more active sites, 2D-layered heterojunction photocatalysts have drawn significant interest and exhibit considerable potential in photocatalysis. Theoretically, by combining the S-scheme heterojunction and gradual 2D hybrid interfaces, one can maximize each one's benefits and improve charge separation and utilization in photocatalysis. The development of sophisticated S-scheme heterojunctions with gradual 2D coupling interfaces for significantly accelerated photocatalytic destruction of refractory organic pollutants, however, has received very little attention to date. ZnO-based heterostructures have been identified as an effective way of increasing photocatalytic efficiency [21–24]. Recently, graphitic carbon nitride (g-$C_3N_4$)-based ZnO nanocomposites have attracted great attention due to their low band gap, high physicochemical/photochemical stability, and electron–hole separation ability [25,26]. In general, semiconductor-mediated photocatalysis begins with the absorption of proper photon energy equal to or greater than the energy gap of the desired photocatalyst to produce $e^-/h^+$ pairs, followed by the production of reactive species that leads to various redox reactions on the heterogeneous photocatalysts' surface [27,28].

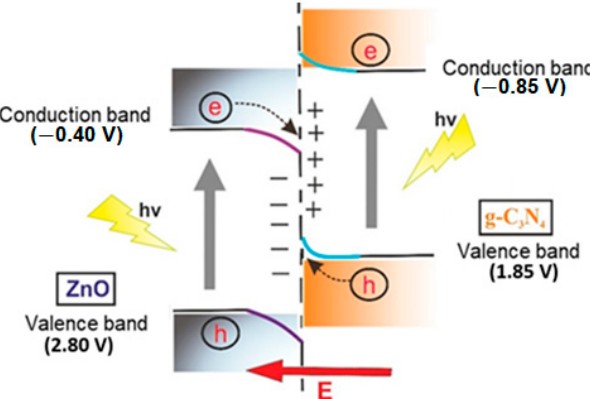

**Figure 1.** A representative diagram for charge-transfer processes in an S-scheme heterojunction under light irradiation.

The latest achievements in the photocatalytic synthesis of organic chemicals using g-$C_3N_4$-based composites have been described by Akhundi et al. [29]. Owing to the intriguing properties of g-$C_3N_4$, including an appropriate bandgap, biocompatibility, cost-effectiveness, and proper stability, g-$C_3N_4$ has attracted widespread attention [30,31]. As a polymeric semiconductor, g-$C_3N_4$ is used in photovoltaic solar cells, chemical sensors, and in separation, detection, bioimaging, photocatalytic, and membrane applications [32]. Accordingly, electron–hole pairs can be generated by light irradiation, which is responsible for initiating the photocatalytic reactions. However, electron–hole pairs can rapidly recombine, which can hamper the degradation efficiency. Therefore, attempts to enhance the

photocatalytic activity of g-C$_3$N$_4$-based photocatalysts have propelled them to the frontline of investigations [33]. In this way, various adjustments, such as doping with efficient elements and preparing nanocomposites, can be performed to decrease the electron and hole recombination by capturing the electrons [34]. Various nanocomposites have been designed by combining g-C$_3$N$_4$ with inorganic materials, including TiO$_2$, CdS, and Fe$_2$O$_3$ [35–38]. Recently, modified g-C$_3$N$_4$ by ZnO has demonstrated remarkable catalytic properties [39]. As an instance, Liu et al. [40] reported a g-C$_3$N$_4$/ZnO nanocomposite structure derived from melamine and zinc chloride with improved photoactivity. As a result, the nanocomposite demonstrated better photocatalytic results compared to bare g-C$_3$N$_4$ or ZnO. In addition, Sun et al. [41] deduced that a g-C$_3$N$_4$/ZnO composite obtained by heating melamine and zinc acetate had a remarkable impact on the photocatalytic degradation of methyl orange and p-nitrophenol under visible light in comparison to the single-phase g-C$_3$N$_4$.

Figure 2 provides the energy band structure of the g-C$_3$N$_4$/ZnO nanocomposite, which demonstrates the lower CB potential energy of g-C$_3$N$_4$ rather than ZnO (ECB = −0.85 eV vs. −0.40 eV). Therefore, by exposing it to visible light, the excited electrons from the CB of g-C$_3$N$_4$ can readily inject into the CB of ZnO. The potential energy match in the g-C$_3$N$_4$/ZnO is a driving force for transferring electrons, which effectively reduces the recombination of e$^-$/h$^+$ pairs. Afterward, the electrons on the CB of ZnO can react with O$_2$ to form superoxide radicals (O$_2$$^{\bullet-}$), which are able to generate hydroxyl radicals (·OH) through further reactions. On the other hand, OH radicals are also produced by the oxidation of water molecules by h+ in the VB of g-C$_3$N$_4$ [42].

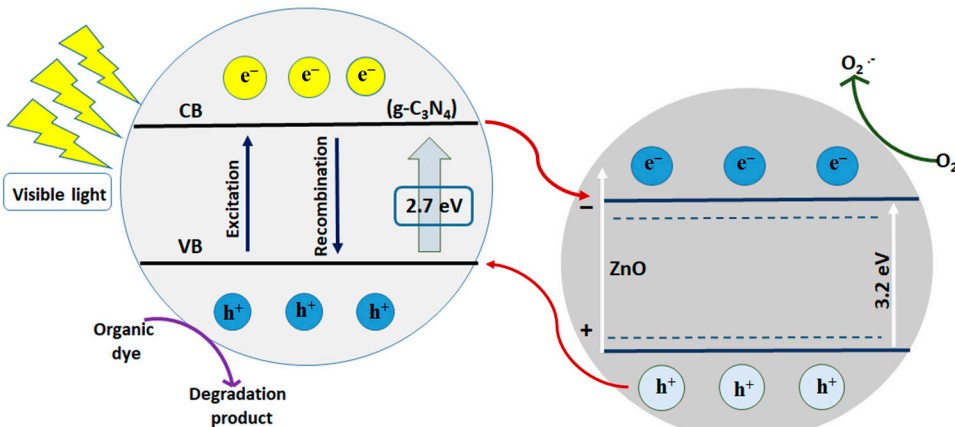

**Figure 2.** A representative diagram for the energy band structure of g-C$_3$N$_4$/ZnO heterostructure.

In this study, g-C$_3$N$_4$/ZnO nanocomposites with different ZnO molar ratios were synthesized using a simple method that is different from the literature. However, the present work is focused on the photocatalytic activity of the g-C$_3$N$_4$/ZnO nanocomposites with different ZnO molar ratios under UV light, which can help understand their performance under real ambient test conditions. To our knowledge, there are no reports on the application of g-C$_3$N$_4$/ZnO nanocomposites for the photocatalytic degradation of crystal violet under UV light. It is observed that the photodegradation of crystal violet is faster and more efficient when the g-C$_3$N$_4$/ZnO nanocomposite is used as a photocatalyst. This is because the recombination rate of photogenerated electron–hole pairs decreases, and the g-C$_3$N$_4$/ZnO nanocomposite increases visible light absorption compared to individual ZnO and g-C$_3$N$_4$.

## 2. Results and Discussion

### 2.1. Characterization of g-C$_3$N$_4$/ZnO Heterogeneous Photocatalyst

The powder X-ray diffraction patterns of the as-prepared bare g-C$_3$N$_4$ and different molar ratios of ZnO-decorated g-C$_3$N$_4$ nanocomposites are shown in Figure 3A. The measured powder XRD patterns of the g-C$_3$N$_4$/ZnO nanocomposites replicate Joint Com-

mittee on Powder Diffraction Standards (JCPDS) card no. 36-1451 and show peaks that are linked to the hexagonal wurtzite structure of ZnO. The XRD patterns of the bare g-C$_3$N$_4$ (JCPDS card no. 87-1526), ZnO, g-C$_3$N$_4$/ZnO (0.25), g-C$_3$N$_4$/ZnO (0.50), and g-C$_3$N$_4$/ZnO (1.00) demonstrate a strong peak at 27.5°, which corresponds to the (002) plane, revealing graphitic stacking of g-C$_3$N$_4$ in the bare g-C$_3$N$_4$ and g-C$_3$N$_4$/ZnO nanocomposites (Figure 3A). By further increasing the molar ratio of ZnO from 0.50 mmol to 1.00 mmol, it can be observed that the crystal growth of the bare g-C$_3$N$_4$ is limited [41]. As the amount of ZnO increases in the samples, the peaks also increase. In addition, the g-C$_3$N$_4$ peaks decrease, which is just opposite of ZnO. This is because the density of g-C$_3$N$_4$ is lower than ZnO (Figure 3A).

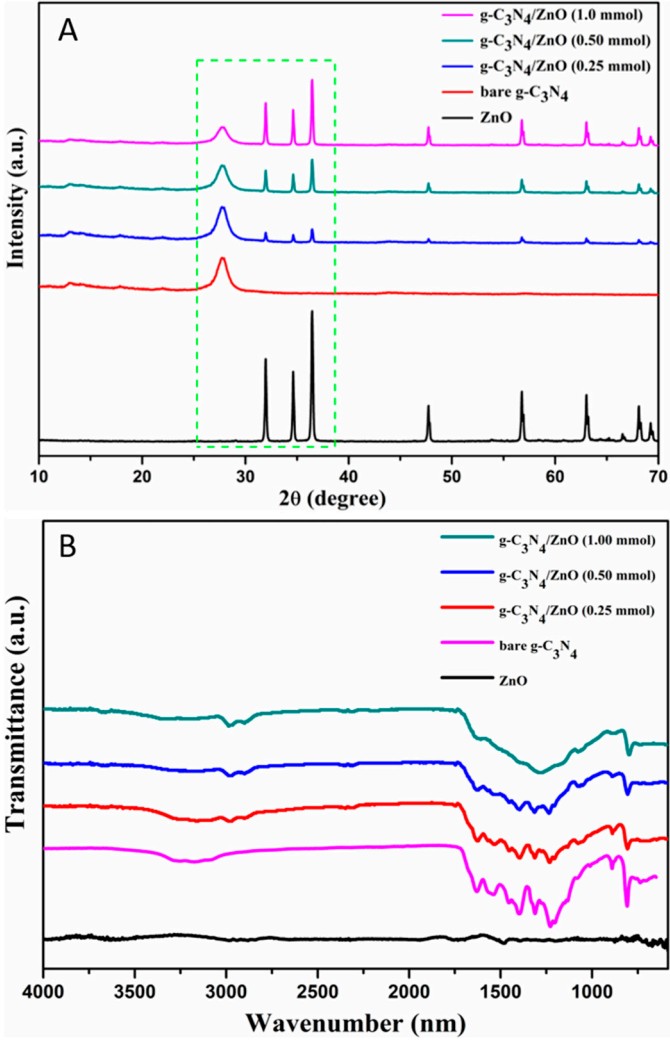

**Figure 3.** (**A**) XRD patterns of the bare g-C$_3$N$_4$, ZnO, and synthesized different molar ratios g-C$_3$N$_4$/ZnO samples, and (**B**) FTIR spectra of the g-C$_3$N$_4$/ZnO samples.

Figure 3B displays the Fourier-transform infrared spectra of the g-C$_3$N$_4$ and g-C$_3$N$_4$/ZnO samples. The N–H and hydroxyl groups of adsorbed H$_2$O molecules give g-C$_3$N$_4$ a broad peak at 3180 cm$^{-1}$. Aromatic C–N stretching modes generated by the g-C$_3$N$_4$ peaks can be assigned to several peaks ranging from 1285 to 1367 cm$^{-1}$. The out-of-plane bending vibration of triazine rings is revealed by the intense band at 806 cm$^{-1}$. The ZnO sample shows characteristic bands in the region of 593–458 cm$^{-1}$, which correspond to the ZnO bond vibration, and another band of about 3160 cm$^{-1}$, which is related to the hydroxyl group stretching vibrations. For the g-C$_3$N$_4$/ZnO hybrid structure, all the characteristic peaks related to g-C$_3$N$_4$ emerge, and the structure is proven to be successfully

synthesized. The typical bands of the ZnO bond vibration appear at about 458, 473, and 455 cm$^{-1}$ for the g-C$_3$N$_4$/ZnO (0.25 mmol), g-C$_3$N$_4$/ZnO (0.50 mmol), and g-C$_3$N$_4$/ZnO (1.00 mmol) samples, respectively. Through ZnO crystallization, the intensities of the peaks at 1626–1206 cm$^{-1}$ drop, and the peaks are merged into a broad absorption band with increasing ZnO content. Hence, the creation of a conjugated material consisting of g-C$_3$N$_4$ and ZnO is vivid in the FT-IR spectra of the g-C$_3$N$_4$/ZnO composites, offering a channel for charge carrier transfer, resulting in photocatalytic activity enhancement [39].

SEM analysis was used to investigate the surface morphology and microstructure of the prepared samples. The SEM images of the bare g-C$_3$N$_4$ (Figure 4A) are in good consistency with the morphology of g-C$_3$N$_4$, which has been previously reported in the literature [42,43]. As illustrated in Figure 4A, the bare g-C$_3$N$_4$ appears as agglomerated pieces containing irregular smaller crystals. Then, a comparison of the surface morphologies of the g-C$_3$N$_4$/ZnO composite structure was performed (Figure 4B–D). The SEM images shown in Figure 4 reveal the formation of ZnO nanoparticles on the g-C$_3$N$_4$ nanosheets. The presented ZnO nanoparticles are not in a globular shape due to the nearness of g-C$_3$N$_4$ sheets. Moreover, Figure 4A shows that the uneven and projecting boundaries of the overlapping graphitic carbon nitride layers are what have created the g-C$_3$N$_4$ multilayer roughness. In Figure 4B–D, we can see the morphological changes of the coating around the g-C$_3$N$_4$ sample as the amount of ZnO increases.

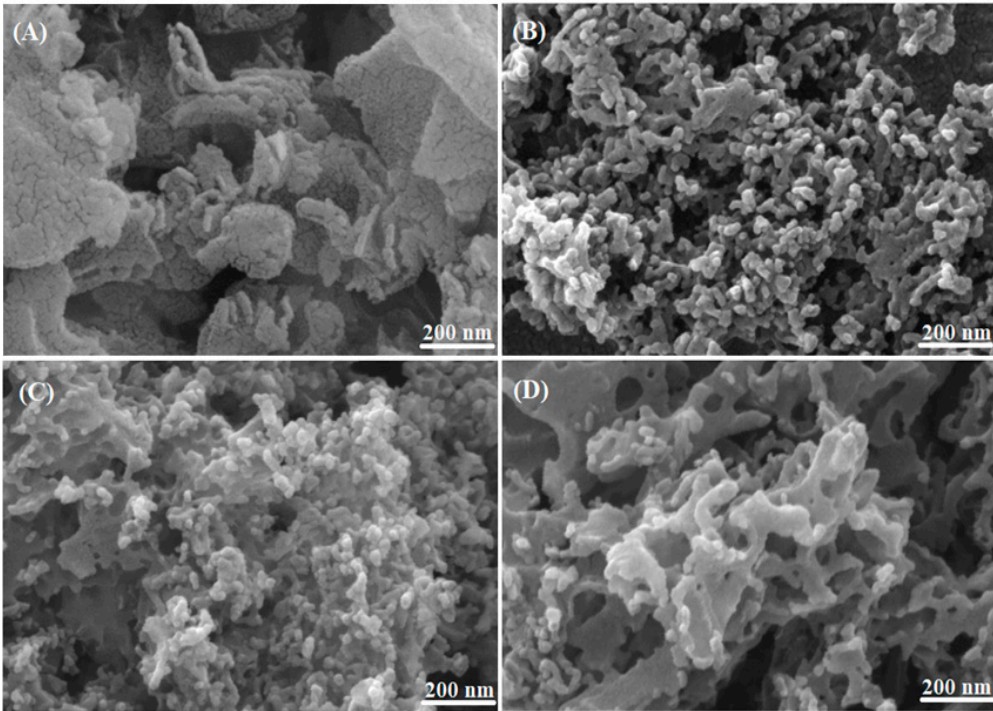

**Figure 4.** SEM images of (**A**) bare g-C$_3$N$_4$, (**B**) g-C$_3$N$_4$/ZnO (0.25), (**C**) g-C$_3$N$_4$/ZnO (0.50), and (**D**) g-C$_3$N$_4$/ZnO (1.00) samples.

The optical properties of the samples were analyzed using a UV–Vis absorption spectroscopy. The normalized absorption spectra of the prepared g-C$_3$N$_4$ and g-C$_3$N$_4$/ZnO nanocomposites are depicted in Figure 5A. The bare g-C$_3$N$_4$ sample has a characteristic absorption peak at 410 nm. We also observe that the absorbance of the g-C$_3$N$_4$/ZnO nanocomposite sample is slightly increased compared to the g-C$_3$N$_4$ sample in the UV–Vis wavelength region. The absorption nature of the g-C$_3$N$_4$/ZnO nanocomposites increases over the entire visible spectrum compared to the bare g-C$_3$N$_4$ and ZnO samples. This improved light absorption intensity in the visible range will be beneficial for the photocatalytic degradation process. The energy bandgap was characterized using the Kubelka–Munk equation for direct band-gap semiconductors, which was derived from a Tauc plot of $(\alpha h\upsilon)^2$

vs. hυ, as shown in Figure 5B. According to Figure 5B, the intersection of the line extrapolation and the x-axis reveals the energy band gaps. The equation is $\alpha = A(h\upsilon - Eg)^n/h\upsilon$, where $\alpha$ is the absorption coefficient, hυ is the energy of the incident photon, Eg is the band gap of the material, A is the constant, and n is a factor that depends on the nature of the electron transition [44]. The band gap energies of the ZnO, bare g-$C_3N_4$, g-$C_3N_4$/ZnO (0.25), g-$C_3N_4$/ZnO (0.50), and g-$C_3N_4$/ZnO (1.00) samples are determined to be 3.10, 2.70, 2.73, 2.66, and 2.61 eV, respectively, as demonstrated in Figure 5B.

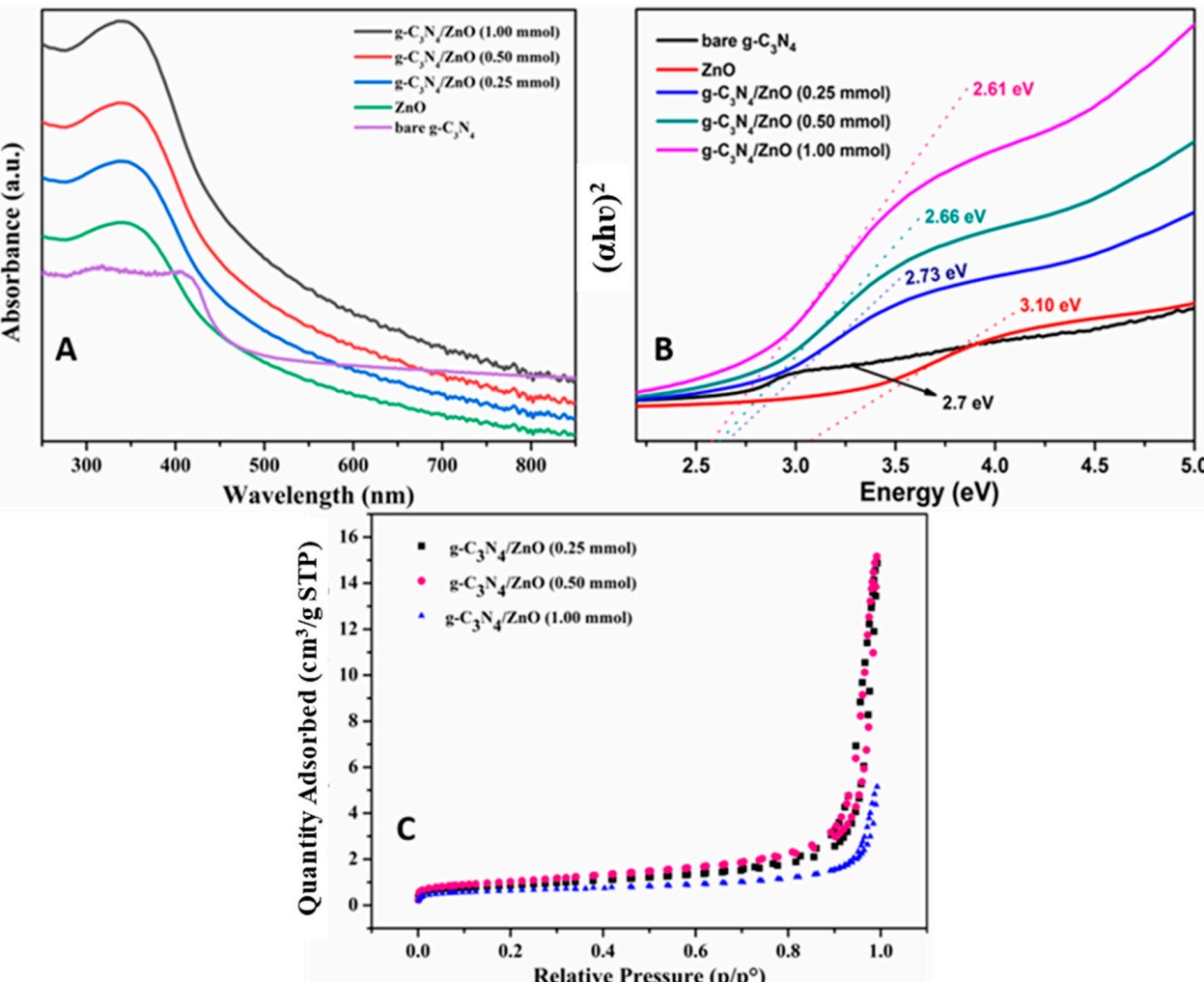

**Figure 5.** (**A**) UV–Vis absorption spectra, (**B**) Tauc plot of the g-$C_3N_4$, ZnO, and different molar ratios g-$C_3N_4$/ZnO nanocomposites, and (**C**) nitrogen adsorption−desorption isotherms of the g-$C_3N_4$/ZnO nanocomposites.

Figure 5C displays the adsorption–desorption $N_2$ isotherms for the g-$C_3N_4$/ZnO hybrid structures. The $N_2$ adsorption–desorption isotherms produced at 77 K were used to determine the surface area and pore volumes of the g-$C_3N_4$/ZnO (0.25 mmol), g-$C_3N_4$/ZnO (0.50 mmol) and g-$C_3N_4$/ZnO (1.00 mmol) hybrid structures (Table 1). The surface areas obtained for the g-$C_3N_4$, g-$C_3N_4$/ZnO (0.25 mmol), g-$C_3N_4$/ZnO (0.50 mmol), and g-$C_3N_4$/ZnO (1.00 mmol) nanocomposites are, respectively, 47.01 $m^2$/g, 67.15 $m^2$/g, 80.77 $m^2$/g, and 49.97 $m^2$/g. The highest surface area and pore volume are obtained for the g-$C_3N_4$/ZnO (0.50 mmol) sample as 80.77 $m^2$/g and 0.38 $cm^3$/g, respectively. The low surface area and pore volume of the hybrid structures can be predicted owing to the changes in the bond structures.

**Table 1.** The surface area and pore volume of the prepared samples.

| Sample | Surface Area (m$^2$/g) | Pore Volume (cm$^3$/g) |
|---|---|---|
| g-C$_3$N$_4$/ZnO (0.25 mmol) | 67.15 | 0.41 |
| g-C$_3$N$_4$/ZnO (0.50 mmol) | 80.77 | 0.38 |
| g-C$_3$N$_4$/ZnO (1.00 mmol) | 49.97 | 0.12 |
| g-C$_3$N$_4$ | 47.01 | 0.17 |

## 2.2. The Effect of Different Amounts of ZnO

To determine the impact of diverse amounts of ZnO on CV degradation efficiency, different amounts of ZnO in the g-C$_3$N$_4$/ZnO photocatalyst were provided. The removal of CV was also investigated for the bare g-C$_3$N$_4$ catalyst to observe the photocatalytic performances of the designed nanocomposites. Initially, the amounts of ZnO selected to explore the degradation efficiency were 0.25, 0.50, and 1.00 mmol. Afterward, 0.10 g/L of the prepared photocatalyst was inserted into the CV aqueous solution, and the photocatalytic decolorization was investigated under UV light irradiation (Figure 6A). The results depict that 0.50 mmol of ZnO can degrade the CV solution up to 95.9% within 120 min. Lower removal efficiencies are obtained for 0.25 mmol and 1.00 mmol of ZnO. The reason could be explained by the fact that lower amounts of ZnO are insufficient to degrade the dye molecules. On the other hand, the active centers of the catalyst might have been inactivated because 1.00 mmol of ZnO was agglomerated due to overdose. Therefore, the 0.50 mmol ZnO-decorated g-C$_3$N$_4$ catalyst was chosen for further evaluation.

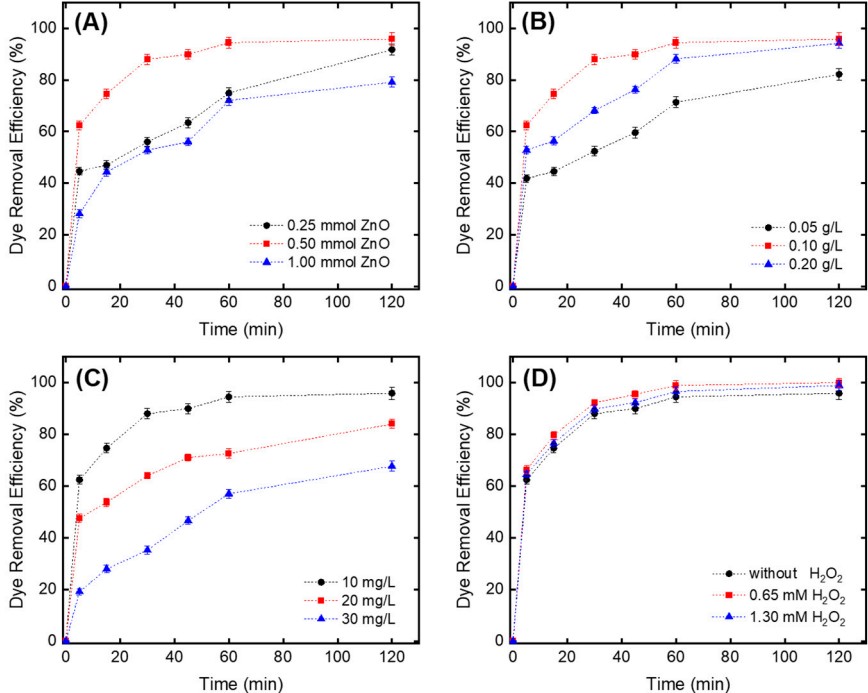

**Figure 6.** (**A**) The effect of ZnO on dye removal efficiency (experimental conditions: [catalyst] = 0.10 g/L). (**B**) The effect of catalyst amount on dye removal efficiency (experimental conditions: ZnO ratio = 0.5 mmol and [dye] = 10 mg/L). (**C**) The effect of dye concentration on dye removal efficiency (experimental conditions: ZnO ratio = 0.5 mmol and [catalyst] = 0.10 g/L). (**D**) The effect of H$_2$O$_2$ concentration on dye removal efficiency (experimental conditions: ZnO ratio = 0.5 mmol, [catalyst] = 0.10 g/L, and [dye] = 10 mg/L).

## 2.3. The Effect of g-C$_3$N$_4$/ZnO Catalyst Concentration

The effect of g-C$_3$N$_4$/ZnO catalyst concentration on dye removal efficiency was examined. Three different catalyst amounts (50, 100, and 200 mg/L) were used to examine the

impact on CV removal efficiency (Figure 6B). The outcomes reveal that by increasing the amount of the catalyst from 0.05 g/L to 0.10 g/L, the degradation efficiency can be increased from 82.2% to 95.9%, respectively. By increasing the catalyst amount, the provided surface area as a platform for the reaction and generation of diverse reactive oxygen species (ROSs) is incremented. However, enhancing the catalyst amount up to 0.20 g/L can decrease the removal efficiency to 94.2%, which might be due to the light penetration prevention.

### 2.4. The Effect of CV Concentration

The influence of initial CV concentration (10, 20, and 30 mg/L) on the removal efficiency is presented in Figure 6C. Increasing the dye concentration from 10 to 30 mg/L causes competition as there are more dye molecules on the constant active sites of the catalyst and the generated ROS [45]. Moreover, high contaminant concentrations cause more intense color and adversely affect dye removal efficiency due to decreasing UV light penetration into the pollutant solution. It seems that most of the UV light might be absorbed by the dye molecules rather than the catalyst [46].

### 2.5. The Effect of $H_2O_2$ Concentration

The effect of $H_2O_2$ concentration on CV removal efficiency is demonstrated in Figure 6D. As it is well known, the addition of $H_2O_2$ promotes the formation of more OH• radicals by reacting with UV light irradiation, and these OH• radicals are able to improve the photocatalytic oxidation [47]. A complete decolorization (100%) of the CV dye was obtained at 0.65 mM of $H_2O_2$ within 120 min. However, an increase in $H_2O_2$ concentration cannot increase the dye removal efficiency due to the prevention of OH• formation by excessive $H_2O_2$ molecules [48].

The efficiency of catalyst reuse was studied by recycling the same catalyst slurry for the degradation of 10 mg/L of CV under similar batch conditions. The reuse experiments were performed for three cycles and by maintaining the catalyst slurry concentration at constant values for each run.

### 2.6. Kinetic Study

The pseudo-first- and second-order reaction kinetics were calculated for the present study. The kinetic constants of the reaction $k_1$ (1/min) and $k_2$ (L/mg.min) were calculated from the slope of the line of the kinetic models of pseudo-first order (ln $C_0/C = k_1 t$) and second order ($1/C - 1/C_0 = k_2 t$). The kinetic constants $k_1$ and $k_2$ and the regression coefficients ($R^2$) are compared in Table 2. The results show that the kinetic constants decrease with an increase in the initial concentration of CV from 10 to 30 mg/L. The pseudo-second-order kinetic model supplies the best fit to the experimental data for photocatalytic decolorization of CV using the g-$C_3N_4$/ZnO catalyst (Figure S1).

**Table 2.** The pseudo-first- and second-order kinetic rate constants with regression coefficients for decolorization of CV at different dye concentration conditions (experimental conditions: solution pH = 6.8, and [catalyst] = 0.10 g/L).

| $C_{CV}$ (mg/L) | First Order | | Second Order | |
|---|---|---|---|---|
| | $k_1$ (1/min) | $R^2$ | $k_2$ (L/mg min) | $R^2$ |
| 10 | 0.0235 | 0.7594 | 0.0202 | 0.9497 |
| 20 | 0.0127 | 0.8297 | 0.0020 | 0.9780 |
| 30 | 0.009 | 0.9290 | 0.0006 | 0.9805 |

### 2.7. The Effect of Different Processes

Adsorption experiments were also performed to compare the efficiency of adsorption and the photocatalytic potential of the g-$C_3N_4$/ZnO catalyst on the removal of the CV dye solution using 0.1 g/L of catalyst loading. The results in Figure 7A indicate that photocatalysis is the dominant mechanism for the decolorization of the CV dye. It is

reported in the literature that the pH of zero-point charge ($pH_{PZC}$) of ZnO nanoparticles is around 9.5–10, which means that for pH values lower than 10, the surface of ZnO nanoparticles is positively charged [49,50]. Additionally, the g-$C_3N_4$ sample has positive zeta potential values [51]. In an aqueous solution, the methyl groups of CV dissociate and then CV molecules are converted to cationic dye ions. Therefore, the surface of the g-$C_3N_4$ and ZnO nanoparticles is positively charged at the original pH (6.8) of the solution, which is disadvantageous for the reach of positively charged CV dye ions due to electrostatic repulsive force. This causes less adsorption of the CV dye on the g-$C_3N_4$/ZnO catalyst. Moreover, the bare g-$C_3N_4$ was also tested for dye decolorization, and a 76.4% dye removal efficiency was obtained. Moreover, a comparative study with the literature is presented in Table 3. The results depict that our heterogeneous catalyst supplies effective decolorization with a lower catalyst amount compared to other studies.

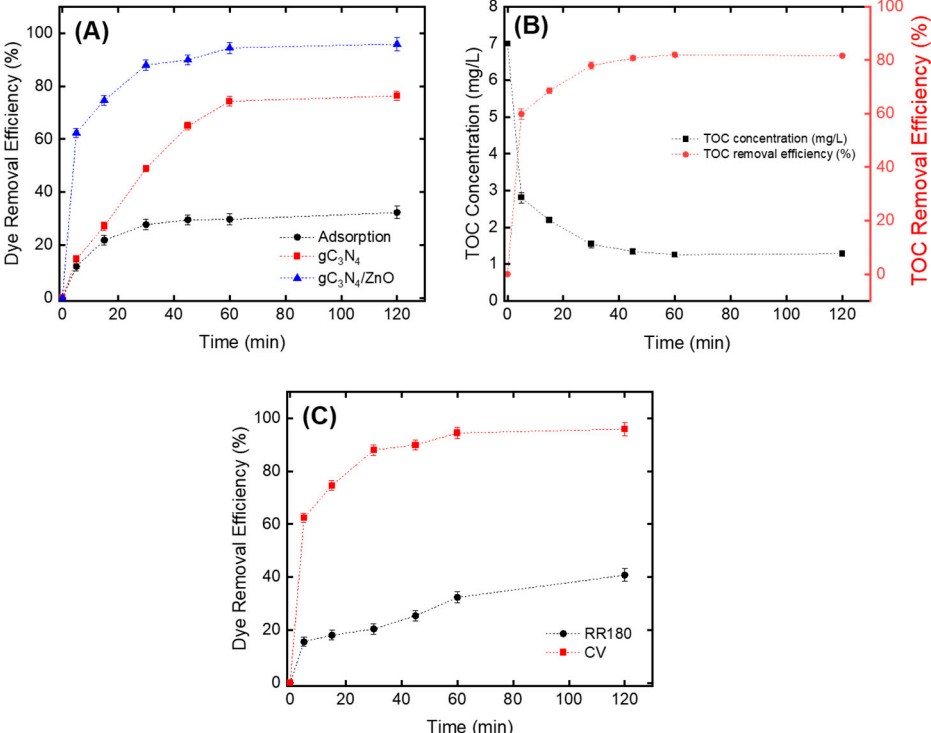

**Figure 7.** (**A**) The effect of different processes on dye removal efficiency, (**B**) TOC alterations during photocatalysis of CV, and (**C**) comparison of the degradation efficiency of RR180 and CV during photocatalysis. (Experimental conditions: ZnO ratio = 0.5 mmol, [catalyst] = 0.10 g/L, and [dye] = 10 mg/L).

**Table 3.** Comparison of g-$C_3N_4$-based heterogeneous catalysts for dye decolorization.

| Catalysts | Dye Concentration (mg/L) | Pollutants | Degradation Efficiency | References |
|---|---|---|---|---|
| g-$C_3N_4$/ZnO | 10 | Crystal Violet | 95.9% for 120 min | This study |
| g-$C_3N_4$/Ag | 10 | Methyl Orange | 98.7% for 120 min | [36] |
| g-$C_3N_4$/CdS | 5 | Rhodamine B | 95% for 120 min | [37] |
| g-$C_3N_4$/BiOBr | 10 | Rhodamine B | 95% for 30 min | [52] |
| g-$C_3N_4$/TiOF$_2$ | 5 | Rhodamine B | 78.3% for 80 min | [53] |
| g-$C_3N_4$/MoS$_2$ | 10 | Rhodamine B | 97.6% for 50 min | [54] |
| g-$C_3N_4$/LaFeO$_3$ | 15 | Rhodamine B | 70% for 160 min | [55] |
| g-$C_3N_4$/Cu$_2$O/Cu | 10 | Methyl Orange | 99% for 40 min | [56] |

The TOC of the solution was monitored during the degradation reactions to ensure the mineralization of the pollutants. It can be clearly seen that the degradation reaches steady-state conditions after 45 min of the process, with 80.12% efficiency (Figure 7B), which indicates that the g-C$_3$N$_4$/ZnO nanocomposite can effectively mineralize CV. After 45 min, degradation is almost not observed. The result show that when the catalyst is new (virgin), the rate of TOC removal is faster, and later its performance remains constant.

### 2.8. The Comparison of Removal Efficiency in the Presence of an Anionic Dye

Additionally, the photocatalytic degradation efficiency of a negatively charged reactive red 180 (RR180) dye was compared with the positively charged CV dye (Figure 7C). Removal efficiencies of 40.8% and 95.9% were obtained for the RR180 and CV dyes within 120 min, respectively. This could be attributed to the molecular structure of the RR180 dye (linear formula: $C_{29}H_{19}N_3Na_4O_{17}S_5$ with molecular weight: 933.76) being more complex than the CV dye (linear formula: $C_{25}H_{30}N_3Cl$ with molecular weight: 407.98).

## 3. Materials and Methods

### 3.1. Materials

Zinc nitrate hexahydrate ($Zn(NO_3)_2 \cdot 6H_2O$, >99%), urea ($CH_4N_2O$ >99.5%), and ammonium hydroxide solution ($NH_4OH$, 26%) were purchased from Sigma-Aldrich (USA). All reagents used in this work were of analytical grade and were used without further purifications. Crystal violet (CV), which was used as a model pollutant, was procured from Sigma-Aldrich (USA).

### 3.2. Synthesis of g-C$_3$N$_4$

Graphite-like g-C$_3$N$_4$ nanosheets were synthesized according to the procedure reported previously in the literature [39]. A total of 50 g of urea was first calcinated at 580 °C for 3 h at a heating rate of 5 °C/min under argon gas flow using a one-step synthesis technique. The material was cooled to room temperature, washed with HNO$_3$ solution (0.1 mol/L) and water, and dried at 70 °C overnight.

### 3.3. The Synthesis of Different Molar Ratios of ZnO-Decorated g-C$_3$N$_4$

The ZnO-decorated g-C$_3$N$_4$ (named g-C$_3$N$_4$/ZnO) nanohybrid was obtained in the presence of urea and different proportions of $Zn(NO_3)_2 \cdot 6H_2O$. Firstly, 10 mL of distilled water was adjusted to a pH of 10 with the ammonium solution. A total of 166 mmol of urea was added to the distilled water/ammonia solution mixture. The urea was completely dissolved, and different proportions of $[Zn(NO_3)_2 \cdot 6H_2O]_x$ (x; 0.25 mmol, 0.50 mmol, and 1.00 mmol) were added to form the hybrid structure. Then, the mixture was vigorously stirred at 70 °C for 2 h, and the solvent was evaporated at 80 °C. The obtained precipitation was heated at 550 °C for 3 h in a furnace (heating rate of 5 °C/min) and then allowed to cool down to room temperature. A schematic illustration of the g-C$_3$N$_4$/ZnO synthesis procedure is shown in Figure 8.

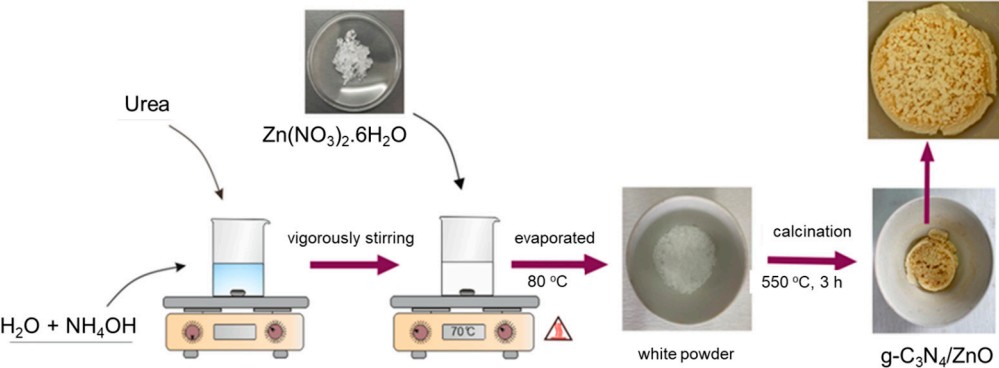

**Figure 8.** The synthesis procedure of different molar ratios of ZnO-decorated g-C$_3$N$_4$.

*3.4. Photocatalytic Degradation Experiments*

The experimental studies were performed in a batch Pyrex photo reactor (column-shaped) with a 500 mL capacity. A total of 100 mL of dye solution was prepared. The photocatalytic reactor was blown from the bottom at a flow rate of 150 mL/min to accelerate the mixing of the catalyst. Cooled air was used to keep a constant temperature of $25 \pm 1$ °C. Six UVA lamps (Philips TL8 W Actinic BL, emitting wavelengths at 365 nm) were placed in a hexagonal position around the reactor, which was surrounded by an aluminum-coated film, to obtain a uniform reflection. The samples were taken from the photocatalytic reactor at predetermined times (5, 15, 30, 45, 60, and 120 min) during the 120 min reaction and centrifuged at 6000 rpm for 5 min to separate the catalyst from the aqueous solution. Maximum absorbance was measured by using a Hach DR3900 UV–Vis spectrophotometer (Düsseldorf, Germany) to calculate dye removal efficiency. The dye removal efficiency was calculated by subtracting the final concentration from the initial concentration and dividing it by the initial concentration.

The reusability behavior of the catalyst was examined through five consecutive runs. After each run, the treated solution was discharged from the reactor after the catalyst was precipitated. Then, a new solution was filled into the reactor and the catalyst was utilized again. The catalyst was washed with 50 mL of deionized water, centrifuged, and dried in an oven.

*3.5. Characterizations Techniques*

The crystal structures of the as-prepared samples were characterized using an X-ray diffraction (XRD, Bruker AXS D8 Advance, Billerica, Massachusetts, USA) diffractometer. The absorption spectra of the g-$C_3N_4$/ZnO nanocomposites were recorded using a UV–Vis spectrophotometer (Thermo scientific-Genesys 150, Menlo Park, CA, USA). A Quanta 650 FEG field emission scanning electron microscope manufactured by FEI (USA) was used to evaluate the surface and section morphology of the nanocomposites. The Fourier-transform infrared spectroscopy (FTIR) spectra of the nanocomposites were recorded using a PerkinElmer FTIR spectrometer (Waltham, MA, USA). The surface area and total pore volume were measured by using the Brunauer Emmett-Teller analysis (BET, MicroActive for TriStar II Plus 2.00, Norcross, GA, USA).

## 4. Conclusions

In summary, the ZnO-decorated g-$C_3N_4$ heterogeneous catalyst was successfully synthesized via a one-step calcination method. It was characterized using FTIR, XRD, UV–Vis, SEM, and BET analyses. The XRD and FTIR measurements demonstrated the strong coordination of ZnO with bare g-$C_3N_4$ and the partial breakdown of the g-$C_3N_4$ catalyst's crystalline structure upon the addition of ZnO. Furthermore, the ZnO-decorated g-$C_3N_4$ composite exhibited high photocatalytic activity for the degradation of CV dye under UV light. The results displayed no decrease in the CV dye removal efficiency up to five cycles for the g-$C_3N_4$ /ZnO photocatalyst. The synergistic interactions between the ZnO and g-$C_3N_4$ provide a forceful method for combining two visible-light photocatalysts to expand their UV light absorption. This study provides some new insights into the fabrication of advanced catalysts for highly efficient photocatalytic applications.

**Supplementary Materials:** The following supporting information can be downloaded at: https://www.mdpi.com/article/10.3390/catal13030485/s1, Figure S1: Kinetic study for CV decoloriaztion.

**Author Contributions:** Conceptualization, B.S. and Z.B.; methodology, T.S.R.; investigation, N.D., K.O. and A.K.; writing—original draft preparation, K.O. and A.K.; writing—review and editing, N.D., K.O. and AK. All authors have read and agreed to the published version of the manuscript.

**Funding:** This research received no external funding.

**Data Availability Statement:** Not applicable.

**Conflicts of Interest:** The authors declare no conflict of interest.

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
