# Peer review of "Preparation of S-Scheme g-C3N4/ZnO Heterojunction Composite for Highly Efficient Photocatalytic Destruction of Refractory Organic Pollutant"

_catalysts, doi:10.3390/catal13030485_

Round 1
Reviewer 1 Report
The manuscript " Design and construction of S-scheme g-C3N4/ZnO heterojunc- 2 tion for highly efficient photocatalytic destruction of refractory 3 organic pollutant" is interesting but not novel. There are a lot of changes required before its publication. My comments are :
1. Much work is already done on g-C3N4/ZnO. What is the novelty of the manuscript? mention in the Introduction section very clearly.
2. What is the efficiency of the system? I can see that 0.10 g/L catalyst in 10 mg/L is too much in the ratio of both.
3. What is the significance of using the same composites for degrading the dye which is done by numerous researchers and also 10 mg/L which is very resulting less efficient? Please explain this.
4. See the advantages of ZnO-based composites for photocatalysis over others and incorporate them in the manuscript. https://doi.org/10.1007/s10311-022-01398-w
5. Please mention the band gap and CB, and VB edge potentials of g-C3N and ZnO in Figure 1.
6. Line 89 TiO2, write correctly
7. Line 102, "Afterward, the electrons on the CB of ZnO can react with O2 to form super- 102 oxide radicals (O2 •Ì¶ ), " Did the authors check the generation of superoxide radicals are thermodynamically possible in between the CB and VB of composites?
8. Figure 4, include an SEM image of ZnO also or give a reference and take an image from the literature.
9. Discussion about Figure 5 (a) needs to improve. mention the specific peaks for all. I can see that all shows peaks in the UV region then how authors claimed the band gap is lower than ZnO. Is it possible?
10. Include error bars in Figure 7 (b)
11. Section 4.4. The authors have not mentioned the reaction in the dark which is adsorption and time.
12. Figure 7 (a) shows the adsorption data separately from other to2. I cannot understand why in others they are done directly in the light. What about adoption desorption equilibrium?
13. Authors should include a UV spectra showing the decrease in dye concentration.
14. Authors should perform TGA to check the stability of composites.
15. The title should be improved " Design and construction look weird.
16. Surprisingly there is no reference to 2022 and 2023. As the manuscript doesn't claim novel work so he work should be compared with other work and recent literature should be cited.
17. Manuscript has a lot of grammatical errors. Authors should improve their English by some native English-speaking person or English improving services.
Author Response
Reviewer #1: The manuscript "e; Design and construction of S-scheme g-C3N4/ZnO heterojunction for highly efficient photocatalytic destruction of refractory 3 organic pollutant" is interesting but not novel. There are a lot of changes required before its publication. My comments are:
Comment 1. Much work is already done on g-C3N4/ZnO. What is the novelty of the manuscript? mention in the Introduction section very clearly.
Response: We thanks to the referee for this comment. We have made the necessary arrangements.
In this study, g-C3N4/ZnO nanocomposites with different ZnO molar ratios were synthesized using a simple method different from the literature. However, present work is focused on the photocatalytic activity of g-C3N4/ZnO nanocomposite with different ZnO molar ratios under UV light, which can help to understand its performance under real ambient test conditions. To our knowledge, there are no reports of the application of g-C3N4/ZnO nanocomposites for the photocatalytic degradation of crystal violet under UV light. It has been observed that the photodegradation of crystal violet is faster and more efficient when the g-C3N4/ZnO nanocomposite is used as a photocatalyst. This is because the recombination rate of photogenerated electron-hole pairs decreases, and the g-C3N4/ZnO nanocomposite increases visible light absorption compared to individual ZnO and g-C3N4.
Comment 2. What is the efficiency of the system? I can see that 0.10 g/L catalyst in 10 mg/L is too much in the ratio of both.
Response: We apologize, but we do not agree with the referee. 0.1 g/L catalyst is quite small amount compared to the literature.
Comment 3. What is the significance of using the same composites for degrading the dye which is done by numerous researchers and also 10 mg/L which is very resulting less efficient? Please explain this.
Response: In this study, g-C3N4/ZnO nanocomposites with different ZnO molar ratios were synthesized using a simple method different from the literature. However, present work is focused on the photocatalytic activity of g-C3N4/ZnO nanocomposite with different ZnO molar ratios under UV light, which can help to understand its performance under real ambient test conditions.
Comment 4. See the advantages of ZnO-based composites for photocatalysis over others and incorporate them in the manuscript. https://doi.org/10.1007/s10311-022-01398-w.
Response: We thanks to the referee for this comment. It was added as ref [14].
Comment 5. Please mention the band gap and CB and VB edge potentials of g-C3N4 and ZnO in Figure 1.
Response: We completed the arrangement by specifying the CB and VB values on the Figure, as in the comment of the referee.
Figure 1. A representative diagram for charge-transfer processes in an S-scheme heterojunction under light irradiation.
Comment 6. Line 89 TiO2, write correctly.
Response: Spelling mistakes have been rechecked and corrected.
Comment 7. Line 102, "Afterward, the electrons on the CB of ZnO can react with O2 to form superoxide radicals (O2•Ì¶ ), "Did the authors check the generation of superoxide radicals are thermodynamically possible in between the CB and VB of composites?
Response: We did not check the generation of superoxide radicals between the CB and VB of composites. However, in the literature it is extensively studied that the generation of superoxide radicals are thermodynamically possible in between the CB and VB of composites. Fig. 2 can be proved this phenomenon [42].
Comment 8. Figure 4, include an SEM image of ZnO also or give a reference and take an image from the literature.
Response: In the present study, g-C3N4 was synthesized in-stu. The synthesis was completed using the starting materials of the ZnO nanomaterial. Therefore, it was not necessary to add the SEM image of the ZnO nanomaterial.
Comment 9. Discussion about Figure 5 (a) needs to improve. mention the specific peaks for all. I can see that all shows peaks in the UV region then how authors claimed the band gap is lower than ZnO. Is it possible?
Response: First of all, we would like to thank the referee for this comment. The incorrect comment in Figure 5a. has been corrected and explained.
The optical properties of the samples were analyzed using UV-Vis absorption spectroscopy. The normalized absorption spectra of the prepared g-C3N4 and g-C3N4/ZnO nanocomposites are depicted in Fig. 5a. The bare g-C3N4 sample has a characteristic absorption peak at 410 nm. We also observed that the absorbance of the g-C3N4/ZnO nanocomposite sample was slightly increased compared to g-C3N4 in the UV-Vis wavelength region. The absorption nature of g-C3N4/ZnO nanocomposites increases over the entire visible spectrum compared to bare g-C3N4 and ZnO. This improved light absorption intensity in the visible range will be beneficial for the photocatalytic degradation process.
Comment 10. Include error bars in Figure 7 (b).
Response: Two different experiments were performed for TOC and no difference was observed between the two readings. Therefore, it was not possible to give the error bars.
Comment 11. Section 4.4. The authors have not mentioned the reaction in the dark which is adsorption and time.
Response: We would like to thank the referee for this comment. We mentioned in Section 2.7
Comment 12. Figure 7 (a) shows the adsorption data separately from other to 2. I cannot understand why in others they are done directly in the light. What about adoption desorption equilibrium?
Response: g-C3N4/ZnO nanocomposite sample was used for both adsorption and photo catalysis. Results in Fig. 7a indicated that photocatalysis was the dominant mechanism for the decolorization of CV dye
Comment 13. Authors should perform TGA to check the stability of composites.
Response: We are sorry but it is impossible to obtain TGA analysis for this study.
Comment 14. The title should be improved "Design and construction look weird.
Response: The title was improved as follows: Preparation of S-scheme g-C3N4/ZnO heterojunction composite for highly efficient photocatalytic destruction of refractory organic pollutant
Comment 15. Surprisingly there is no reference to 2022 and 2023. As the manuscript doesn't claim novel work so he works should be compared with other work and recent literature should be cited.
Response: Thank you to the referees for their feedback. Citations have been rechecked and current publications have been added to the references.
Comment 16. Manuscript has a lot of grammatical errors. Authors should improve their English by some native English-speaking person or English improving services.
Response: For grammatical errors in the manuscript, the article was again read and required changes were made in the sentence.
Reviewer 2 Report
Title: Design and construction of S-scheme g-C3N4/ZnO heterojunction for highly efficient photocatalytic destruction of refractory organic pollutant
This study with the facile preparation of S-scheme g-C3N4/ZnO heterojunction for catalytic activity is interesting. Besides this, there are still some concerns which need to be addressed before its acceptance for publishable. Please see the comments below thoroughly.
1. Abstract is not written in well-organized form. Author has claimed degradation is dominant over adsorption, however comparative data between them has not provided.
2. Where are the fitting graphs for kinetic studies; pseudo first and second order? Please provide the graphs. Also provide significance of kinetic study i.e. pseudo second order which was the best fitted for your experiment.
3. Experiments at different conditions in terms of pH and initial concentrations should be studied.
4. Characteristics of the material is studied in superficial way. More discussion and systematic way of writing for characterization should be needed.
5. More characterizations (TEM and XPS) and discussion in details should be done.
6. Correct/double check the formulas of all compounds (e.g., TiO2 not TiO2). Regression coefficient (R2), not R2. Authors provided the data for TOC but missed discussion about it.
7. To insight/elaborate of the degradation phenomenon from various material designations as well as technologies, you can prefer the following literatures as follows.
Chemosphere, Volume 309, Part 1, December 2022, 136638; http://dx.doi.org/10.1016/j.chemosphere.2022.136638 , Chemical Engineering Journal, Volume 417, 1 August 2021, 129312, Catalysts 2020, 10(5), 546; https://doi.org/10.3390/catal10050546
8. Section 4 must be conclusion NOT discussion. So, it would be better to give section 2 as Results and Discussion.
Author Response
Reviewer #2: Title: Design and construction of S-scheme g-C3N4/ZnO heterojunction for highly efficient photocatalytic destruction of refractory organic pollutant. This study with the facile preparation of S-scheme g-C3N4/ZnO heterojunction for catalytic activity is interesting. Besides this, there are still some concerns which need to be addressed before its acceptance for publishable. Please see the comments below thoroughly.
Comment 1. Abstract is not written in well-organized form. Author has claimed degradation is dominant over adsorption, however comparative data between them has not provided.
Response: Abstract is re-organized. Moreover, comparative data was added for adsorption.
Comment 2. Where are the fitting graphs for kinetic studies; pseudo first and second order? Please provide the graphs. Also provide significance of kinetic study i.e. pseudo second order which was the best fitted for your experiment.
Response: Kinetic fit plots (2.6) were presented as supplementary information.
Comment 3. Experiments at different conditions in terms of pH and initial concentrations should be studied.
Response: We thank the referee for suggestion. We studied initial concentration between 10-30 mg/L (Fig. 6c). However, we did not observe a big difference for solution pH between 4-10.
Comment 4. Characteristics of the material is studied in superficial way. More discussion and systematic way of writing for characterization should be needed
Response: We thank the referee for suggestion. Some comments were added to UV-Vis and BET analyzes in characterization interpretations.
Comment 5. More characterizations (TEM and XPS) and discussion in details should be done.
Response: We are sorry but it is impossible to obtain TEM and XPS analyses for this study due to lack of the project budget.
Comment 6. Correct/double check the formulas of all compounds (e.g., TiO2 not TiO2). Regression coefficient (R2), not R2. Authors provided the data for TOC but missed discussion about it.
Response: Corrected types in the article. More discussion for TOC was added.
Comment 7. To insight/elaborate of the degradation phenomenon from various material designations as well as technologies, you can prefer the following literatures as follows.
Chemosphere, Volume 309, Part 1, December 2022, 136638; http://dx.doi.org/10.1016/j.chemosphere.2022.136638 , Chemical Engineering Journal, Volume 417, 1 August 2021, 129312, Catalysts 2020, 10(5), 546; https://doi.org/10.3390/catal10050546
Response: We thank the referee for suggestion.
Comment 8. Section 4 must be conclusion NOT discussion. So, it would be better to give section 2 as Results and Discussion.
Response: The order of the article was revised upon the reviewer's comment.
- Results and Discussion, 3. Materials and Methods, 4. Conclusion
Reviewer 3 Report
The authors have successfully prepared a g-C3N4/ZnO photocatalyst starting from urea, which was used for the photocatalytic degradation of CV under UV light. Basic structural studies were performed, from which conclusions need to be corrected. I was not convinced that adsorption was sufficiently taken into account in the photodegradation experiments.
My questions and comments:
1. I disagree that ZnO is more stable and has a smaller band gap than TiO2 (line 47)
2. Figures 1 and 2 contradict each other. The CB and VB levels are good in Figure 2. Figure 1 and the corresponding paragraph should be reconsidered! I would suggest deleting it completely.
3. The use of the subscript has been omitted in several cases: Lines: 47, 89, 124, 184, 185, 246, 255, 256, and in the list of references.
4. The discussion of UV-Vis spectra is unclear. How are the spectra normalised? The mentioned absorption peak at 410 nm is rather an absorption edge. "Two characteristic absorption peaks" was mentioned, but I can see only one.
5. A reference is needed for the Tauc method.
6. The symbols should not be connected by a solid line in Figures 6 and 7.
7. What is the error of the dye removal efficiency (DRE)? Considering the trend and the scatter of points, it is probably larger than indicated.
8. Absorption spectral series for the curves shown in Figures 6 and 7 and kinetic fit plots (2.6) should be presented as supplementary information.
9. The meaning of „ROS” should be given in the line 233.
10. Figure 7.a shows that after 15 minutes, more than 20% of the CV is adsorbed. How much can this distort the DRE results? How is it possible that the DRE shown is actually the sum of the degraded and adsorbed CV?
11. How does adsorption affect the kinetic fit?
12. What is the explanation for the TOC removal efficiency being greater than the DRE after 5 to 15 minutes?
13. Did you carry out a blank measurement with UV illumination but without a catalyst?
14. In Table 3, it would be more correct to report the best result without H2O2. Data cannot be compared without specifying quntum efficiency. It would be useful to determine the absorbed photons by the catalyst.
15. Why do the authors only mention the investigation of the reusability of the catalyst in the "Discussion" section? This should be presented in chapter 2.
16. The list of references must be checked! I found more than 50 typos (indexes, empty brackets "()", ...)
Author Response
Reviewer #3: The authors have successfully prepared a g-C3N4/ZnO photocatalyst starting from urea, which was used for the photocatalytic degradation of CV under UV light. Basic structural studies were performed, from which conclusions need to be corrected. I was not convinced that adsorption was sufficiently taken into account in the photodegradation experiments.
My questions and comments:
Comment 1. I disagree that ZnO is more stable and has a smaller band gap than TiO2 (line 47)
Response: The type in the introduction was corrected upon the referee's warning and completed as in the explanation below. We thank the referee for this warning.
ZnO is an n-type semiconductor oxide comparable to TiO2. As ZnO has the same band gap energy as TiO2 but exhibits greater absorption efficiency across a significant portion of the solar spectrum, ZnO has been suggested as a potential replacement photocatalyst.
Comment 2. Figures 1 and 2 contradict each other. The CB and VB levels are good in Figure 2. Figure 1 and the corresponding paragraph should be reconsidered! I would suggest deleting it completely.
Response: We think that this graphic is important in terms of emphasizing the importance of the study. Other referees agree with this view.
Figure 1. A representative diagram for charge-transfer processes in an S-scheme heterojunction under light irradiation.
Comment 3. The use of the subscript has been omitted in several cases: Lines: 47, 89, 124, 184, 185, 246, 255, 256, and in the list of references.
Response: The types in the current study have been corrected.
Comment 4. The discussion of UV-Vis spectra is unclear. How are the spectra normalized? The mentioned absorption peak at 410 nm is rather an absorption edge. "Two characteristic absorption peaks" was mentioned, but I can see only one.
Response: Fig 5a was rechecked in line with the referee's warnings. The "Two characteristic absorption peaks" interpretations mentioned are corrected as in the explanation below.
The optical properties of the samples were analyzed using UV-Vis absorption spectroscopy. The normalized absorption spectra of the prepared g-C3N4 and g-C3N4/ZnO nanocomposites are depicted in Fig. 5a. The bare g-C3N4 sample has a characteristic absorption peak at 410 nm. We also observed that the absorbance of the g-C3N4/ZnO nanocomposite sample was slightly increased compared to g-C3N4 in the UV-Vis wavelength region. The absorption nature of g-C3N4/ZnO nanocomposites increases over the entire visible spectrum compared to bare g-C3N4 and ZnO. This improved light absorption intensity in the visible range will be beneficial for the photocatalytic degradation process.
Comment 5. A reference is needed for the Tauc method.
Response: https://dx.doi.org/10.1021/acsomega.9b02688 references for Tauc method.
Comment 6. The symbols should not be connected by a solid line in Figures 6 and 7.
Response: Dash line was used.
Comment 7. What is the error of the dye removal efficiency (DRE)? Considering the trend and the scatter of points, it is probably larger than indicated.
Response: We supplied the error bars of the dye removal efficiency. It was not larger than indicated.
Comment 8. Absorption spectral series for the curves shown in Figures 6 and 7 and kinetic fit plots (2.6) should be presented as supplementary information.
Response: Kinetic fit plots (2.6) were presented as supplementary information.
Comment 9. The meaning of “ROS” should be given in the line 233.
Response: The meaning of ROS was added as follows: reactive oxygen species (ROSs)
Comment 10. Figure 7.a shows that after 15 minutes, more than 20% of the CV is adsorbed. How much can this distort the DRE results? How is it possible that the DRE shown is actually the sum of the degraded and adsorbed CV?
Response: We thank to the reviewer for this comment. However, we are sorry but we disagree with the referee. The DRE is not the sum of the degraded and adsorbed CV.
Comment 11. How does adsorption affect the kinetic fit?
Response: We did not study for this study.
Comment 12. What is the explanation for the TOC removal efficiency being greater than the DRE after 5 to 15 minutes?
Response: We are grateful to the referee for her/his attention. The experiments were repeated and the new graphs were placed.
Comment 13. Did you carry out a blank measurement with UV illumination but without a catalyst?
Response: Yes, we did it. No dye removal was observed with UV illumination without a catalyst.
Comment 14. In Table 3, it would be more correct to report the best result without H2O2. Data cannot be compared without specifying quantum efficiency. It would be useful to determine the absorbed photons by the catalyst.
Response: We thank the referee for suggestion. Ignoring H2O2, 95.9% is reported in Table 3.
Comment 15. Why do the authors only mention the investigation of the reusability of the catalyst in the "Discussion" section? This should be presented in chapter 2.
Response: We thank the referee for suggestion. The following paragraph was added in Section 2.
The efficiency of catalyst reuse was studied by recycling the same catalyst slurry for degradation of 10 mg/L CV under similar batch conditions. The reuse experiments were performed for about 3 cycles and by maintaining the catalyst slurry concentration at con-stant values for each run.
Comment 16. The list of references must be checked! I found more than 50 typos (indexes, empty brackets "()", ...)
Response: The list of references was checked carefully.
Reviewer 4 Report
In this work, graphitic carbon nitride (g-C3N4)-based ZnO heterostructure was synthe sized by a facile calcination method utilizing urea and zinc nitrate hexahydrate as initiators. According to the scanning electron microscopy (SEM) images, spherical ZnO particles can be seen along 17 with the g-C3N4 nanosheets. Additionally, X-Ray diffraction (XRD) analysis revealed the successful synthesis of the g-C3N4/ZnO. The photocatalytic activity of the synthesized catalyst was tested for the decolorization of crystal violet (CV) as an organic refractory contaminant. The impact of ZnO molar ratio, catalyst amount, CV concentration, and H2O2 concentration were investigated on CV degradation efficiency. The manuscript can be reconsidered for publication in Catalysts after addressing the comments:
1. It would be more interesting if the authors focus more on the significance of their findings in the abstract.
2. Please make sure your conclusions' section underscores the scientific value-added of your paper, and/or the applicability of your results. Highlight the novelty of your study. Clearly discuss what the previous studies that you are referring to are.
3. fig.5 c, y axis should be cm3/g (in case of gaseous compounds!). alpha coeefecient is also missed in y-axis of fig.5b.
4. Table 1, SBET and pore volume of bared C3N4 should be added to make logical comparison.
5. Table 2 should be edited, remove columns 2 and 3 with adding explanation related to pH and catalyst dosage.
6. 30 % of adsorption capacity should be considered in reporting the real photocatalytic removal efficiency over the prepared photocatalysts, both text and conclusion section.
7. Fig. 7 a, the content should be clarified, dark tests for bared C3N4, best as-prepared S-scheme photocatalyst besides photolysis test should be well reported.
8. new photocatalytic materials based on C3N4 and ZnO can be specifically considered in introduction section:
https://www.sciencedirect.com/science/article/abs/pii/S0925838820333193
https://www.sciencedirect.com/science/article/abs/pii/S2352186420315121
https://pubs.acs.org/doi/abs/10.1021/acs.langmuir.2c01822
Author Response
Reviewer 4#: In this work, graphitic carbon nitride (g-C3N4)-based ZnO heterostructure was synthesized by a facile calcination method utilizing urea and zinc nitrate hexahydrate as initiators. According to the scanning electron microscopy (SEM) images, spherical ZnO particles can be seen along 17 with the g-C3N4 nanosheets. Additionally, X-Ray diffraction (XRD) analysis revealed the successful synthesis of the g-C3N4/ZnO. The photocatalytic activity of the synthesized catalyst was tested for the decolorization of crystal violet (CV) as an organic refractory contaminant. The impact of ZnO molar ratio, catalyst amount, CV concentration, and H2O2 concentration were investigated on CV degradation efficiency. The manuscript can be reconsidered for publication in Catalysts after addressing the comments:
Comment 1. It would be more interesting if the authors focus more on the significance of their findings in the abstract.
Response: The abstract was reorganized.
Comment 2. Please make sure your conclusions' section underscores the scientific value-added of your paper, and/or the applicability of your results. Highlight the novelty of your study. Clearly discuss what the previous studies that you are referring to are.
Response: We discussed as suggested.
Comment 3. Fig. 5c, y axis should be cm3/g (in case of gaseous compounds!). alpha coefficient is also missed in y-axis of Fig. 5b.
Response: We thank the referee for his comment. They were corrected.
Comment 4. Table 1, SBET and pore volume of bared C3N4 should be added to make logical comparison.
Response: We thank the referee for his comment. We detail the information required for the BET analysis of g-C3N4 in Table 1.
Table 1. The surface area and pore volume of the prepared samples.
Sample |
Surface area (m2/g) |
Pore volume (cm3/g) |
g-C3N4/ZnO (0.25 mmol) |
67.15 |
0.41 |
g-C3N4/ZnO (0.50 mmol) |
80.77 |
0.38 |
g-C3N4/ZnO (1.00 mmol) |
49.97 |
0.12 |
g-C3N4 |
47.01 |
0.17 |
Comment 5. Table 2 should be edited, remove columns 2 and 3 with adding explanation related to pH and catalyst dosage.
Response: It was corrected as suggested.
Comment 6. 30 % of adsorption capacity should be considered in reporting the real photocatalytic removal efficiency over the prepared photocatalysts, both text and conclusion section.
Response: It was considered in the abstract and text.
Comment 7. Fig. 7a, the content should be clarified, dark tests for bared C3N4, best as-prepared S-scheme photocatalyst besides photolysis test should be well reported.
Response: It was clarified as suggested.
Comment 8. new photocatalytic materials based on C3N4 and ZnO can be specifically considered in introduction section:
https://www.sciencedirect.com/science/article/abs/pii/S0925838820333193
https://www.sciencedirect.com/science/article/abs/pii/S2352186420315121
https://pubs.acs.org/doi/abs/10.1021/acs.langmuir.2c01822
Response: The introductory part of the present study has been edited by citing different and recent articles.
Reviewer 5 Report
The manuscript, titled with “Design and construction of S-scheme g-C3N4/ZnO heterojunction for highly efficient photocatalytic destruction of refractory organic pollutant”, mainly researched the preparation by a calcination procedure and photocatalytic evaluation of g-C3N4/ZnO composites. Actually, to construct composites g-C3N4/ZnO for photocatalyic degradation was a quite old topic and could be found everywhere. In addition, no any new information was supported in this work. Therefore, I had to suggest rejecting such manuscript for following aspects. 1) No relevant photocatalytic degradation by bare C3N4 and ZnO was provided to show the catalytic enhancement; 2) The regular photocatalytic reactions should be set after adsorption saturation; 3) The explanation of photocatalytic improvement should be supported by experimental results; 4) Authors should do some work to support the “S-scheme” of such composites; 5) The variation of catalytic degradation of CV and RR180 might be the adsorption of these dye molecules with different charge over catalysts surface; 6) Relevant references should be updated;
Author Response
Reviewer 5#: The manuscript, titled with “Design and construction of S-scheme g-C3N4/ZnO heterojunction for highly efficient photocatalytic destruction of refractory organic pollutant”, mainly researched the preparation by a calcination procedure and photocatalytic evaluation of g-C3N4/ZnO composites. Actually, to construct composites g-C3N4/ZnO for photocatalyic degradation was a quite old topic and could be found everywhere. In addition, no any new information was supported in this work. Therefore, I had to suggest rejecting such manuscript for following aspects.
Comment 1. No relevant photocatalytic degradation by bare C3N4 and ZnO was provided to show the catalytic enhancement.
Response: Please look at the Fig. 7a in which we provided photocatalytic degradation of bare C3N4.
Comment 2. The regular photocatalytic reactions should be set after adsorption saturation;
Response: All the photocatalytic experiments was carried out after adsorption saturation with dark conditions. After then, lamps were turned on.
Comment 3. The explanation of photocatalytic improvement should be supported by experimental results.
Response: The explanation of photocatalytic improvement was supported by experimental results.
Comment 4. Authors should do some work to support the “S-scheme” of such composites;
Response: Direct Z-scheme heterojunction and S-scheme heterojunction both use the same basic carrier separation mechanism. Traditional and all-solid-state heterojunctions, the first two generations of direct Z-scheme heterojunction, have clear flaws. As a result, S-scheme heterojunction with clear identification and a practical charge-transfer channel can successfully prevent needless misunderstandings. In addition, due to their distinctive structure and photoelectronic properties, such as greater specific surface area, more active sites, and 2D layered heterojunction photocatalysts have drawn significant interest and exhibit considerable potential in photocatalysis. Theoretically, by combining the S-scheme heterojunction and gradual 2D hybrid interfaces, one can maximize each one's benefits and improve charge separation and utilization in photocatalysis. The development of sophisticated S-scheme heterojunctions with cascade 2D coupling interfaces for significantly accelerated photocatalytic destruction of refractory organic pollutant, however, has received very little attention to date.
Comment 5. The variation of catalytic degradation of CV and RR180 might be the adsorption of these dye molecules with different charge over catalysts surface.
Response: The adsorption efficiency of CV was shown in Fig. 7A. Therefore, photocatalytic degradation is the main mechanism.
Comment 6. Relevant references should be updated.
Response: In the current study, the references have been changed and updated.
Round 2
Reviewer 1 Report
The manuscript can be accepted in present form
Author Response
We thank to Reviewer.
Reviewer 3 Report
2nd review of manuscript ID: catalysts-2114519
Although the authors have corrected several mistakes, unfortunately new errors have been introduced in the manuscript and I can only partially accept the answers. The experimental conditions are still not clear, and without clarification I cannot recommend the manuscript for acceptance.
My replies to the authors' responses:
I accept answers 6, 9, 14 and 15.
The authors have written in line 47-48: „Moreover, ZnO has unique features such as abundance, low toxicity, ecofriendly and cost-effectiveness [14].”
Although the cited [14] review article really does contain this sentence, the original article cited in that review uses a slightly different wording. Comparing the MSDS sheets for ZnO and TiO2, it is clear that ZnO is highly toxic („H410: Very toxic to aquatic life with long lasting effects”), while TiO2 is not.
My original comment 2. „Figures 1 and 2 contradict each other. The CB and VB levels are good in Figure 2. Figure 1 and the corresponding paragraph should be reconsidered!”
The modified Figure 1 has been supplemented with the values of the CB and VB potentials, which – apart from being incorrect ("−0.40 V" negative sign missing) – clearly show that the figure is wrong. The energy levels are plotted incorrectly. Probably the right side shows g-C3N4 and the left side shows ZnO. Accordingly, the figure is in contradiction with the corresponding text (lines 74-83).
My original comment 3: „The use of the subscript has been omitted in several cases: Lines: 47, 89, 124, 184, 185, 246, 255, 256, and in the list of references.”
In the original 255 and 256 lines (now 279-280), the formulas are still incorrect (the correct formula would be „ln(C0/C) = k1t” and „1/C – 1/C0 = k2t”) and the reference list has not been fully corrected. The subscripts have been omitted everywhere.
The authors did not answer the question of how the absorption spectra were normalized. In Figure 5, the spectra of pure ZnO should decrease much more sharply between 360-400 nm ("White powder")
Ref. [39] is not the most relevant article for Tauc method. The original Tauc work should be referred: https://doi.org/10.1002/pssb.19660150224
I would have asked for the whole absorption spectra series (comment 8) because it would show the possible intermediates, which would be good for interpreting the TOC results.
Figure captions are missing for the suppl. figures. The axis captions are incorrect: the data suggest that the left figure is likely to be ln(C/C0) and the right figure 1/C -1/C0. In this case, however, the fact that the intercepts of fitted lines are not equal to zero (at t = 0 min) suggests that the authors did not wait for the absorption equilibrium before illumination (comment 11). The very sharp initial slopes of the curves in Figure 6 suggest the same. The value of k in neither a first nor a second order process can depend on the initial concentration! What are the reasons for the measured differences? Heterogeneous photocatalysis is much more complex than to simplify its kinetics so much.
In Figs 6 and 7, it needs to be clearly shown how the CV concentration changed before starting the illumination (adsorption period). The authors still did not specify how long they waited for the adsorption equilibrium to set up. I was not convinced about the correct method of performing their measurement.
If the authors have performed a blank test (illumination without a catalyst), this should be included in the article.
The reference list still contains many errors (omitted subscripts, reference [39] and [44] are identical, the latter is not referenced in the text). Although the authors did not rewrite the part of the introduction concerning references [1] to [12] (lines 33-47), they replaced references 3, 6, 7, 9 and 12. Why? As I see it, the originals were more relevant.
In the revised Figure 1, the potentials for g-C3N4 and ZnO CB are shown as -0.85 V and -0.40 V, respectively. In the text (lines 116 and 117) the same parameters are -1.1 V and -0.27 V. Somehow this discrepancy should be resolved.
Author Response
Reviewer #3: Although the authors have corrected several mistakes, unfortunately new errors have been introduced in the manuscript and I can only partially accept the answers. The experimental conditions are still not clear, and without clarification I cannot recommend the manuscript for acceptance.
Response: We are grateful to the Reviewer.
My replies to the authors' responses:
Comment 1. The authors have written in line 47-48: „Moreover, ZnO has unique features such as abundance, low toxicity, ecofriendly and cost-effectiveness [14].”
Although the cited [14] review article really does contain this sentence, the original article cited in that review uses a slightly different wording. Comparing the MSDS sheets for ZnO and TiO2, it is clear that ZnO is highly toxic („H410: Very toxic to aquatic life with long lasting effects”), while TiO2 is not.
Response: In the cited article; https://doi.org/10.1016/j.matchemphys.2018.04.053
“Presently, ZnO is considered as a good semiconductor photocatalyst because of its excellent absorption of ultraviolet (UV) light, suitable electronic band structure, low cost, and nontoxicity. However, the photogenerated electron and hole pairs easily produce excitons due to their strong coulombic interaction, and the rapid recombination of excitons significantly weakens the photocatalytic activity of ZnO”. The statement is included.
Comment 2. My original comment 2. „Figures 1 and 2 contradict each other. The CB and VB levels are good in Figure 2. Figure 1 and the corresponding paragraph should be reconsidered!”
The modified Figure 1 has been supplemented with the values of the CB and VB potentials, which – apart from being incorrect ("−0.40 V" negative sign missing) – clearly show that the figure is wrong. The energy levels are plotted incorrectly. Probably the right side shows g-C3N4 and the left side shows ZnO. Accordingly, the figure is in contradiction with the corresponding text (lines 74-83).
Response: Thanks for your comment. CB and VB band values have been corrected again. In addition, the energy levels were also changed. Figures 1 and 2 are arranged to be compatible.
Figure 1. A representative diagram for charge-transfer processes in an S-scheme heterojunction under light irradiation.
Comment 3. My original comment 3: „The use of the subscript has been omitted in several cases: Lines: 47, 89, 124, 184, 185, 246, 255, 256, and in the list of references.”
In the original 255 and 256 lines (now 279-280), the formulas are still incorrect (the correct formula would be „ln(C0/C) = k1t” and „1/C – 1/C0 = k2t”) and the reference list has not been fully corrected. The subscripts have been omitted everywhere.
Response: Subscripts have been rechecked. The desired arrangements are completed as follows.
The kinetic constants of the reaction k1 (1/min) and k2 (L/mg.min) were calculated from the slope of the line of the kinetic models of pseudo-first order (ln C0/C = k1t) and sec-ond-order (1/C − 1/ C0 = k2t).
Comment 4. The authors did not answer the question of how the absorption spectra were normalized. In Figure 5, the spectra of pure ZnO should decrease much more sharply between 360-400 nm ("White powder")
Response: The absorption spectra taken in Figure 5 were obtained by taking solid UV. Normalization is not done. The same measurement results were found with the reference article;
https://dx.doi.org/10.1021/acsomega.9b02688.
Comment 5. Ref. [39] is not the most relevant article for Tauc method. The original Tauc work should be referred: https://doi.org/10.1002/pssb.19660150224.
Response: Thank you for your suggestion. Added requested ref for Tauc method.
[44] Tauc, J., Grigorovici, R. and Vancu, A. (1966), Optical Properties and Electronic Structure of Amorphous Germanium. phys. stat. sol. (b), 15: 627-637. https://doi.org/10.1002/pssb.19660150224.
Comment 7. I would have asked for the whole absorption spectra series (comment 8) because it would show the possible intermediates, which would be good for interpreting the TOC results.
Response: We are sorry, but we will not be able to provide for this study
Comment 8. Figure captions are missing for the suppl. figures. The axis captions are incorrect: the data suggest that the left figure is likely to be ln(C/C0) and the right figure 1/C -1/C0. In this case, however, the fact that the intercepts of fitted lines are not equal to zero (at t = 0 min) suggests that the authors did not wait for the absorption equilibrium before illumination (comment 11). The very sharp initial slopes of the curves in Figure 6 suggest the same. The value of k in neither a first nor a second-order process can depend on the initial concentration! What are the reasons for the measured differences? Heterogeneous photocatalysis is much more complex than simplifying its kinetics so much.
Response: The kinetic constants of the reaction k1 (1/min) and k2 (L/mg.min) were calculated from the slope of the line of the kinetic models of pseudo-first order (ln C0/C = k1t) and second-order (1/C − 1/ C0 = k2t).
Comment 10. In Figs 6 and 7, it needs to be clearly shown how the CV concentration changed before starting the illumination (adsorption period). The authors still did not specify how long they waited for the adsorption equilibrium to set up. I was not convinced about the correct method of performing their measurement.
If the authors have performed a blank test (illumination without a catalyst), this should be included in the article.
Response: Adsorption time is 120 minutes. In the absence of light and catalyst only, there was no removal of the CV dye.
Comment 11. The reference list still contains many errors (omitted subscripts, reference [39] and [44] are identical, and the latter is not referenced in the text). Although the authors did not rewrite the part of the introduction concerning references [1] to [12] (lines 33-47), they replaced references 3, 6, 7, 9 and 12. Why? As I see it, the originals were more relevant.
Response: References have been reviewed. References [39] and [44] have been changed. Differences in other references have been corrected with the recommendations of the reviewers.
[39] Li, Y.; Zhu, S.; Liang, Y.; Li, Z.; Wu, S; Chang, C.; Luo, S.; Cui, Z. Synthesis of α- Fe2O3/g-C3N4 Photocatalyst for high-efficiency water splitting under full light. Materials & Design 2020, 109191–. doi:10.1016/j.matdes.2020.109191.
[44] Tauc, J.; Grigorovici, R.; Vancu, A.; Optical Properties and Electronic Structure of Amorphous Germanium. Phys. stat. sol. (b) 1966, 15, 627-637. https://doi.org/10.1002/pssb.19660150224.
Comment 12. In the revised Figure 1, the potentials for g-C3N4 and ZnO CB are shown as -0.85 V and -0.40 V, respectively. In the text (lines 116 and 117) the same parameters are -1.1 V and -0.27 V. Somehow this discrepancy should be resolved.
Response: The mentioned error has been corrected. ‘Fig. 2 provides the energy band structures of the g-C3N4/ZnO nanocomposite, which demonstrates the lower CB potential energy of g-C3N4 rather than ZnO (ECB = -0.85 eV vs. -0.40 eV).
Reviewer 4 Report
1. The language of the manuscript still needs improvement.
2. In section 3.4, authors talked about reusability tests and the coresponding procedure but no as-related results are inserted in the results-discussion section. Please check both sections to solve this concern.
3. comment 8 in the previous report (citing the papers) is ignored by authors. (new photocatalytic materials based on C3N4 and ZnO can be specifically considered in introduction section:
https://www.sciencedirect.com/science/article/abs/pii/S0925838820333193
https://www.sciencedirect.com/science/article/abs/pii/S2352186420315121
https://pubs.acs.org/doi/abs/10.1021/acs.langmuir.2c01822)
4. Scientific values of the key results have not been added to the conclusion section.
good luck
Author Response
Comment 1. The language of the manuscript still needs improvement.
Response: The writing language in the introduction was revised and some adjustments were made.
Comment 2. In section 3.4, authors talked about reusability tests and the corresponding procedure but no as-related results are inserted in the results-discussion section. Please check both sections to solve this concern.
Response: It was added.
Comment 3. comment 8 in the previous report (citing the papers) is ignored by authors. (new photocatalytic materials based on C3N4 and ZnO can be specifically considered in introduction section:
https://www.sciencedirect.com/science/article/abs/pii/S0925838820333193
https://www.sciencedirect.com/science/article/abs/pii/S2352186420315121
https://pubs.acs.org/doi/abs/10.1021/acs.langmuir.2c01822)
Response: Thank you for your suggestion. The three references mentioned have been added to the relevant fields.
Padervand M.; Rhimi B.; Wang C.; One-pot synthesis of novel ternary Fe3N/Fe2O3/C3N4 photocatalyst for efficient removal of rhodamine B and CO2 reduction. Journal of Alloys and Compounds, 2021, 852, 156955. https://doi.org/10.1016/j.jallcom.2020.156955.
Padervand M.; Heidarpour H.; Goshadehzehn M.; Hajiahmadi S.; Photocatalytic degradation of 3-methyl-4-nitrophenol over Ag/AgCl-decorated/[MOYI]-coated/ZnO nanostructures: Material characterization, photocatalytic performance, and in-vivo toxicity assessment of the photoproducts, Environmental Technology & Innovation, 2021, 21, 101212. https://doi.org/10.1016/j.eti.2020.101212.
Forouzandeh-Malati M.; and Ganjali F.; Zamiri E.; and Zarei-Shokat S.; Jalali, F.; and Padervand M.; Taheri-Ledari, R.; Maleki, A.; Efficient Photodegradation of Eriochrome Black-T by a Trimetallic Magnetic Self-Synthesized Nanophotocatalyst Based on Zn/Au/Fe-Embedded Poly(vinyl alcohol), Langmuir, 2022, 38, 13728-13743. 10.1021/acs.langmuir.2c01822.
Comment 4. Scientific values of the key results have not been added to the conclusion section.
Response: It was added.
Reviewer 5 Report
The work can be accepted in current form
Author Response
We thank to Reviewer